

# Quantitative trait locus mapping analysis of multiple traits when using genotype data with potential errors

Liang Tong[1,2], Ying Zhou[3], Yixing Guo[4], Hui Ding[2] and Donghai Ji[1]

[1] School of Science, Harbin University of Science and Technology, Harbin, P. R. China
[2] School of Information Engineering, Suihua University, Suihua, P. R. China
[3] School of Mathematical Sciences, Heilongjiang University and Heilongjiang Provincial Key Laboratory of the Theory and Computation of Complex Systems, Harbin, P. R. China
[4] Dalian University of Science and Technology, Dalian, P. R. China

## ABSTRACT

**Background:** Quantitative trait locus (QTL) analysis aims to locate and estimate the effects of the genes influencing quantitative traits and infer the relationship between gene variants and changes in phenotypic characteristics using statistical methods. Some methods have been developed to map QTLs of multiple traits in the case of no genotype error in a given dataset. However, practical genetic data that people use may contain some potential errors because of the limitations of biotechnology. Common genetic data correction methods can only reduce errors, but cannot calculate the degree of error. In this paper, we propose a QTL mapping strategy for multiple traits in the presence of genotype errors.

**Methods:** The additive effect, dominant effect, recombination rate, error rate, and other parameters of QTLs can be simultaneously obtained using this new method in the framework of multiple-interval mapping.

**Results:** Our simulation results show that the accuracy of parameter estimation can be improved by considering the errors of marker genotypes during the analysis of genetic data. Real data analysis also shows that the new method proposed in this paper can map the QTLs of multiple traits more accurately.

## INTRODUCTION

Recently, multiple-trait gene mapping has been widely studied, which refers to analyzing multiple quantitative/qualitative traits simultaneously, considering the correlations among multiple traits. It is necessary to consider the correlations among multiple traits in the process of quantitative trait locus (QTL) mapping because the joint information of multiple traits can improve the accuracy of detecting QTLs that influence the traits of interest. Multiple-trait methods have been validated to be more powerful than single-trait methods (*Zhu & Zhang, 2009*), and they are being used in many fields such as complex trait analysis and animal or plant genetic breeding.

Corresponding authors
Ying Zhou, zhouying@hlju.edu.cn
Donghai Ji, jidonghai@hrbust.edu.cn

Genome-wide association studies (GWAS) have become a powerful strategy for exploring candidate genes for important traits, including human complex diseases (*Li & Leal, 2008*; *Maity, Sullivan & Tzeng, 2012*; *Sainani, 2015*; *Shi et al., 2017*; *Storey & Tibshirani, 2003*; *Wang et al., 2016*; *Zhang et al., 2017*). Currently, several methods for detecting associations between multiple traits and common/rare variants have been proposed. For instance, *Zhu, Jiang & Zhang (2012)* presented a covariate-adjusted association method based on generalized Kendall' tau, *Zhu & Zhang (2013)* proposed a nonparametric regression method for multiple longitudinal phenotypes using multivariate adaptive splines, *Zhou et al. (2016)* developed a nonparametric method to test for associations between rare variants and multiple traits, *Kwak & Pan (2017)* proposed gene-and pathway-based association tests for multiple traits with GWAS summary statistics, and *Gai & Eskin (2018)* presented a meta-analysis method for multiple-trait association analysis.

In addition, linkage analysis methods, such as interval mapping (IM), composite interval mapping (CIM), and multiple-interval mapping (MIM), have also been used to detect QTLs that control single/multiple traits. Linkage mapping methods have their own advantages in cases of both single and multiple traits and different mapping methods depend on their own statistical models. Unfortunately, however, the linkage analysis method for a single trait cannot accurately map the genes that have multiple effects. *Jiang & Zeng (1995)* proposed the MT-CIM method based on a mixed linear model and maximum likelihood method. This method enables the simultaneous mapping of multiple traits using composite interval mapping. Compared with the single-trait QTL mapping method, it can improve the accuracy and efficiency of QTL mapping. Based on the idea of MIM, using the MT-CIM model, *Joehanes (2009)* proposed a multi-trait and multi-interval mapping method abbreviated MTMIM and showed that the accuracy of MTMIM is higher than that of MIM and MT-CIM. Recently, *Tong, Sun & Zhou (2018)* proposed a method (MTMIM-NEW) for simultaneously estimating QTL parameters for mapping multiple traits, which showed higher precision.

All the above-mentioned methods are applicable to cases where there are no genotype errors included in the practical data set. However, due to the constraints in genotype scoring software and biochemical anomalies, most of the data that researchers have used may contain certain potential genotype errors. The genotype errors cannot be neglected arbitrarily in statistical analysis and may have a significant impact on the study of genetic linkage analysis (*Ronin et al., 2015*; *Tong et al., 2015*; *Yan et al., 2016*).

To circumvent this difficulty, in this paper, we proposed a multi-trait multi-interval mapping method in the presence of genotype errors in the genetic data. The closed iteration formulas of QTL effects, QTL recombination rates, the covariance matrix, and the genotype error rate are given in the new method, which guarantees the precision of parameter estimation. To validate the feasibility of the new method, we analyzed mouse high-density lipoprotein cholesterol data with missing genotypes. The results show that the method can effectively solve the problem of incomplete marker genotypes caused by biological or physical deficiencies. Simulation results also show that the new method has advantages in estimating all parameters of interest when marker genotypes contain errors.

**Table 1 The conditional probabilities of QTL genotypes given the marker genotypes.**

| Code | Marker genotype | QTL genotype | | |
|---|---|---|---|---|
| | | $QQ$ | $Qq$ | $qq$ |
| 1 | $M_iM_iM_{i+1}M_{i+1}$ | 1 | 0 | 0 |
| 2 | $M_iM_iM_{i+1}m_{i+1}$ | $1-r_i$ | $r_i$ | 0 |
| 3 | $M_iM_im_{i+1}m_{i+1}$ | $(1-r_i)^2$ | $2r_i(1-r_i)$ | $r_i^2$ |
| 4 | $M_im_iM_{i+1}M_{i+1}$ | $r_i$ | $1-r_i$ | 0 |
| 5 | $M_im_iM_{i+1}m_{i+1}$ | 0 | 1 | 0 |
| 6 | $M_im_im_{i+1}m_{i+1}$ | 0 | $1-r_i$ | $r_i$ |
| 7 | $m_im_iM_{i+1}M_{i+1}$ | $r_i^2$ | $2r_i(1-r_i)$ | $(1-r_i)^2$ |
| 8 | $m_im_iM_{i+1}m_{i+1}$ | 0 | $r_i$ | $1-r_i$ |
| 9 | $m_im_im_{i+1}m_{i+1}$ | 0 | 0 | 1 |

**Note:**
$r_i = \gamma_{i1}/\gamma_i$.

In particular, compared with existing methods, the new method can provide higher precision in estimating QTL positions (recombination rates).

## THEORY AND METHODS

We consider the data of $n$ individuals from the $F_2$ population with $t$ phenotypic traits; $q + 1$ marker loci are closely linked to form $q$ marker intervals. Assuming that $Y_{jl}$ ($j = 1, \ldots, n, l = 1, \ldots, t$) represents the $l^{th}$ phenotypic trait value of the $j^{th}$ individual, $X_{ji}$ ($j = 1, \ldots, n, i = 1, \ldots, q + 1$) and $\widetilde{X}_{ji}$ ($j = 1, \ldots, n, i = 1, \ldots, q + 1$) denote the true marker genotype and the observed marker genotype, possibly with error of the $i^{th}$ marker of the $j^{th}$ individual, respectively, $X_{ji}^*$ ($j = 1, \ldots, n, i = 1, \ldots, q$) denotes the latent QTL genotype within the $i^{th}$ marker interval of the $j^{th}$ individual. Let $Y_j = (Y_{j1}, Y_{j2}, \ldots, Y_{jt})'$, $X_j = (X_{j1}, X_{j2}, \ldots, X_{j(q+1)})'$, $\widetilde{X}_j = (\widetilde{X}_{j1}, \widetilde{X}_{j2}, \ldots, \widetilde{X}_{j(q+1)})'$ and $X_j^* = (X_{j1}^*, X_{j2}^*, \ldots, X_{jq}^*)'$. $\gamma_i$ and $\gamma_{i1}$ respectively denote recombination rate of the $i^{th}$ marker interval which is known and the recombination rate between the $i^{th}$ marker and the latent QTL in the marker interval. Assuming that each marker interval of the $F_2$ group has at most one QTL (*Zhou, 2010*), and when the genotype combination $X_{ji}^M$ of the $i^{th}$ marker interval of the $j^{th}$ individual is known, the conditional probabilities $P(X_{ji}^*|X_{ji}^M)$ of QTL genotype $X_{ji}^*(Q_iQ_i, Q_iq_i, q_iq_i)$ are presented in Table 1.

Here, we assume that parameter $\theta = P(\widetilde{X}_{ji} \neq x | X_{ji} = x)$ denotes the parameter of the genotype error rate for any genotype $x$, and further assume that $\varphi^j$ denotes the joint error rate of the $j^{th}$ individual. During each step of the iteration computation in our new method, when the true marker genotype $X_j$ is given, we can calculate the number $k_j$ of false genes coded at the $q + 1$ marker loci. Assuming that whether a marker genotype has a genotyping error or not is independent of other marker genotypes, we can obtain

$$\varphi^j = P(\widetilde{X}_j|X_j) = \prod_{i=1}^{q+1} P(\widetilde{X}_{ji}|X_{ji}) = (\theta/2)^{k_j} \cdot (1-\theta)^{(q+1)-k_j}.$$

The joint error rates $\varphi^j (j = 1, \ldots, n)$ can be used to infer the parameter $\theta$ of the genotype error rate in the new method.
## Statistical model

In the framework of multiple-interval mapping, to solve the problem of multiple-trait analysis with genotype errors, we consider building a statistical model as follows:

$$\underset{n\times t}{Y} = \sum_{i=1}^{q}\left[\underset{n\times 1}{u_i}\ \underset{1\times t}{a_i} + \underset{n\times 1}{v_i}\ \underset{1\times t}{d_i}\right] + \underset{n\times t}{e}, \tag{1}$$

where $\underset{n\times t}{Y}$ is the phenotype value matrix with element $Y_{ji}$, $\underset{n\times 1}{u_i} = [u_{1i}, u_{2i}, ..., u_{ni}]'$ and $\underset{n\times 1}{v_i} = [v_{1i}, v_{2i}, ..., v_{ni}]'$ denotes the indicator vector of the $i^{th}$ QTL genotypes of $n$ individuals, respectively. In detail,

$$u_{ji} = 1, \ v_{ji} = -0.5, \quad if\ X_{ji}^* = Q_i Q_i;$$

$$u_{ji} = 0, \ v_{ji} = 0.5, \quad if\ X_{ji}^* = Q_i q_i;$$

$$u_{ji} = -1, \ v_{ji} = -0.5, \quad if\ X_{ji}^* = q_i q_i.$$

$a_i = (a_{i1}, a_{i2}, ..., a_{it})$ and $d_i = (d_{i1}, d_{i2}, ..., d_{it})$ denote the additive effect vector and dominant effect vector of the $i^{th}$ QTL on $t$ traits, respectively.

Furthermore, let the QTL effect matrix be

$$C = \begin{pmatrix} a_{11} & a_{12} & \cdots & a_{1t} \\ d_{11} & d_{12} & \cdots & d_{1t} \\ \vdots & \vdots & & \vdots \\ a_{q1} & a_{q2} & \cdots & a_{qt} \\ d_{q1} & d_{q2} & \cdots & d_{qt} \end{pmatrix},$$

$e = \{e_{ji}\}_{n\times t}$ represents the residual matrix, $e_j = [e_{j1}, e_{j2}..., e_{jt}]'$, and $E(e_j) = 0$, $cov(e_j) = \sum_e$. For convenience, we denote the covariance matrix as:

$$\sum_e = \begin{pmatrix} \sigma_{11} & \sigma_{12} & \cdots & \sigma_{1t} \\ \sigma_{21} & \sigma_{22} & \cdots & \sigma_{2t} \\ \vdots & \vdots & & \vdots \\ \sigma_{t1} & \sigma_{t2} & \cdots & \sigma_{tt} \end{pmatrix},$$

where $\sigma_{il} = cov(e_{ji}, e_{jl}) = \rho\sqrt{\sigma_{ii}\sigma_{ll}}$, $i, l = 1, ..., t$, for any individual, $j$.

To solve the problem of multiple-trait, multiple-interval mapping with genotype errors, based on Model (1), we next provide a detailed derivation process of the new method *via* the classical expectation-maximization (EM) algorithm (*Dempster, Laird & Rubin, 1977*). Let $\Omega = (C, \Sigma_e, \gamma, \theta)$ denote all parameters of interest, where $\gamma = (\gamma_{11}, \gamma_{21}, ..., \gamma_{q1})$ is the parameter vector of QTL positions. The new method has the advantage of simultaneously estimating all model parameters in the framework of multiple-interval mapping.

## Parameter estimation *via* the EM algorithm

In our new method, we focus on the observed data $(\widetilde{X}, Y)$, where $\widetilde{X} = (\widetilde{X}_1, \widetilde{X}_2, ...., \widetilde{X}_n)'$, $Y = (Y_1, Y_2, ...., Y_n)'$, and deeply mine the generating mechanism of $\widetilde{X}$. At the same time,
we need to comply with the statistical model consisting of multiple traits and the latent QTLs given in Model (1). Therefore, based on the logistic structure of all random variables, as well as the background of genetic linkage, the EM algorithm can be applied to infer all parameters of interest here.

The complete log-likelihood function of the parameter matrix $\Omega$ can be expressed as follows:

$$
\begin{aligned}
l(\Omega) &= \log[\prod_{j=1}^{n} P(Y_j|X_j^*, \Omega) \cdot P(X_j^*|X_j, \Omega) \cdot P(\widetilde{X}_j|X_j, \Omega)] \\
&= \sum_{j=1}^{n}[\log P(Y_j|X_j^*, \Omega) + \log P(X_j^*|X_j, \Omega) + \log P(\widetilde{X}_j|X_j, \Omega)].
\end{aligned}
$$

**E−Step**: Given the observed data $(\widetilde{X}, Y)$ and the $s^{th}$-step parameter values $\Omega^{(s)}$, we calculate the conditional expectation $Q(\Omega|\widetilde{X}, Y, \Omega^{(s)})$ of $l(\Omega)$.

$$
\begin{aligned}
Q(\Omega|\widetilde{X}, Y, \Omega^{(s)}) = &\sum_{j=1}^{n}\sum_{X_j^*}\log P(Y_j|X_j^*, \Omega) \cdot \tau_{X_j^*}^{(s)} + \sum_{j=1}^{n}\sum_{X_j}\sum_{X_j^*}\log P(X_j^*|X_j, \Omega) \cdot \tau_{X_j X_j^*}^{(s)} \\
&+ \sum_{j=1}^{n}\sum_{X_j}\log P(\widetilde{X}_j|X_j, \Omega) \cdot \tau_{X_j}^{(s)}.
\end{aligned}
\tag{2}
$$

Here, $\tau_{x_j x_j^*}^{(s)} = P(X_j^* = x_j^*, X_j = x_j|\widetilde{X}_j, Y_j, \Omega^{(s)})$, $\tau_{x_j^*}^{(s)} = P(X_j^* = x_j^*|\widetilde{X}_j, Y_j, \Omega^{(s)})$, and $\tau_{x_j}^{(s)} = P(X_j = x_j|\widetilde{X}_j, Y_j, \Omega^{(s)})$. According to the Bayes formula, $\tau_{x_j x_j^*}^{(s)}$ can be calculated and expressed in the following form:

$$
\tau_{x_j x_j^*}^{(s)} = \frac{P(Y_j|x_j^*, \Omega^{(s)}) \cdot P(x_j^*|x_j, \Omega^{(s)}) \cdot P(\widetilde{X}_j|x_j, \Omega^{(s)})}{\sum_{x_j}\sum_{x_j^*} P(Y_j|x_j^*, \Omega^{(s)}) \cdot P(x_j^*|x_j, \Omega^{(s)}) \cdot P(\widetilde{X}_j|x_j, \Omega^{(s)})}.
$$

Further, we let

$$
\begin{aligned}
\tau_{x_{jk}^*}^{(s)} &= P(X_j^* = x_j^{*k}|\widetilde{X}_j, Y_j, \Omega^{(s)}) \\
&= \sum_{x_j}\frac{P(Y_j|x_j^{*k}, \Omega^{(s)}) \cdot P(x_j^{*k}|X_j, \Omega^{(s)}) \cdot P(\widetilde{X}_j|X_j, \Omega^{(s)})}{\sum_{k=1}^{3^q} P(Y_j|x_j^{*k}, \Omega^{(s)}) \cdot P(x_j^{*k}|X_j, \Omega^{(s)}) \cdot P(\widetilde{X}_j|X_j, \Omega^{(s)})}, \quad \forall k \in \{1, ..., 3^q\}.
\end{aligned}
\tag{3}
$$

where $\tau_{x_{jk}^*}^{(s)}$ represents the conditional probability that $x_j^*$ takes the $k^{th}$ value $x_j^*$ under the given condition of $\widetilde{X}_j, Y_j, \Omega^{(s)}$. In a detailed analysis, $P(Y_j|x_j^{*k}, \Omega^{(s)}) = f(Y_j; \mu_{jk}, \sum_e)$, which is a multivariate normal density with mean $\mu_{jk}$ and covariance matrix $\sum_e$. $P(x_j^{*k}|X_j, \Omega^{(s)})$ can be expressed by a function of conditional probabilities listed in Table 1, because each QTL genotype is conditionally independent, given the condition of marker genotypes. In detail, $P(x_j^{*k}|X_j, \Omega^{(s)})$ is the function of recombination rates $\gamma_{11}, \gamma_{21}, ...,$ and $\gamma_{q1}$.

**M−Step**: Calculate the maximum value of the conditional expectation $Q(\Omega|\widetilde{X}, Y, \Omega^{(s)})$, in order to obtain $\Omega^{(s+1)}$.

### Updating C and $\sum_e$

Based on the first part of $Q(\Omega|\widetilde{\mathbf{X}}, \mathbf{Y}, \Omega^{(s)})$ in Eq. (2), the iteration expression of the QTL effect matrix $C$ and covariance matrix $\sum_e$ can be obtained as follows:

$$C_{cq \times t}^{(s+1)} = R^{(s)} - M^{(s)} C^{(s)},$$

$$\sum_e^{(s+1)} = \frac{1}{n}(\mathbf{Y} - W^{(s)} DC^{(s+1)})'(\mathbf{Y} - W^{(s)} DC^{(s+1)}),$$

where $W = [\tau_{x_{jk}^*}]$ is a $n \times 3^q$ matrix with element $\tau_{x_{jk}^*}$ given in Eq. (3), $G$ is the QTL genotype sequence, and $D = (D_1, D_2, D_3, \ldots, D_{cq})$ is a $3^q \times cq$ matrix with expression

$$D = \begin{bmatrix} 1 & -1/2 & 1 & -1/2 & \cdots & 1 & -1/2 & 1 & -1/2 \\ 1 & -1/2 & 1 & -1/2 & \cdots & 1 & -1/2 & 0 & 1/2 \\ & & & & \vdots & & & & \\ -1 & -1/2 & -1 & -1/2 & \cdots & -1 & -1/2 & 0 & 1/2 \\ -1 & -1/2 & -1 & -1/2 & \cdots & -1 & -1/2 & -1 & -1/2 \end{bmatrix} = \begin{bmatrix} G_{11\cdots11} \\ G_{11\cdots10} \\ \vdots \\ G_{-1-1\cdots-10} \\ G_{-1-1\cdots-1-1} \end{bmatrix};$$

The formulae of $R^{(s)}$ and $M^{(s)}$ can be expressed as

$$R^{(s)} = \begin{bmatrix} \dfrac{D_1' W'^{(s)} Y}{\mathbf{1}' W^{(s)}(D_1 \# D_1)} \\ \vdots \\ \dfrac{D_{cq}' W'^{(s)} Y}{\mathbf{1}' W^{(s)}(D_{cq} \# D_{cq})} \end{bmatrix}_{cq \times t}, \quad M^{(s)} = \begin{bmatrix} 0 & \dfrac{\mathbf{1}' W^{(s)}(D_1 \# D_2)}{\mathbf{1}' W^{(s)}(D_1 \# D_1)} & \cdots & \dfrac{\mathbf{1}' W^{(s)}(D_1 \# D_{cq})}{\mathbf{1}' W^{(s)}(D_1 \# D_1)} \\ \dfrac{\mathbf{1}' W^{(s)}(D_2 \# D_1)}{\mathbf{1}' W^{(s)}(D_2 \# D_2)} & 0 & \cdots & \dfrac{\mathbf{1}' W^{(s)}(D_2 \# D_{cq})}{\mathbf{1}' W^{(s)}(D_2 \# D_2)} \\ \vdots & \vdots & \ddots & \vdots \\ \dfrac{\mathbf{1}' W^{(s)}(D_{cq} \# D_1)}{\mathbf{1}' W^{(s)}(D_{cq} \# D_{cq})} & \dfrac{\mathbf{1}' W^{(s)}(D_{cq} \# D_2)}{\mathbf{1}' W^{(s)}(D_{cq} \# D_{cq})} & \cdots & 0 \end{bmatrix}_{cq \times cq},$$

where notation "#" in expressions $R^{(s)}$ and $M^{(s)}$ denotes the Hadamard product of two vectors and 1 is an $n \times 1$ column vector of ones.

### Updating $\gamma$

We define that the indicator function, $I_{st}^{ji}$, of the QTL genotype and the marker-interval genotype have the following form:

$$I_{(ji)}^{st} = \begin{cases} 1, & X_{ji}^* = s, X_{ji}^M = t, s = 0, 1, 2; t = 1, \ldots, 9, \\ 0, & else. \end{cases}$$

Here, $j = 1, \ldots, n$, $i = 1, \ldots, q$. According to the conditional independence assumption of QTL genotypes, given marker-interval genotypes, the second part of $Q(\Omega|\widetilde{\mathbf{X}}, \mathbf{Y}, \Omega^{(s)})$ in Eq. (2) can be transformed into the following form:

$$\sum_{j=1}^{n} \sum_{X_j} \sum_{X_j^*} \log P(X_j^*|X_j, \Omega) \cdot \tau_{X_j X_j^*}^{(s)} = \sum_{j=1}^{n} \sum_{X_j} \sum_{X_j^*} \tau_{X_j X_j^*}^{(s)} \cdot \sum_{i=1}^{q} \log P(X_{ji}^*|X_{ji}^M, \Omega). \tag{4}$$

Hereinto, $P(X_{ji}^*|X_{ji}^M, \Omega)$ can be expressed as a function of the probabilities listed in Table 1 and the indicator function $I^{st}{}_{(ji)}$s. When we maximize the above Eq. (4), an explicit closed expression of the recombination rate $\gamma_{i1}$ can be obtained:

$$\gamma_{i1}^{(s+1)} = \frac{\gamma_i \cdot \sum_{j=1}^n \sum_{x_j} \sum_{x_j^*} \tau_{x_j x_j^*}^{(s)} \cdot (I_{(04)}^{ji} + 2I_{(07)}^{ji} + I_{(12)}^{ji} + I_{(18)}^{ji} + 2I_{(23)}^{ji} + I_{(26)}^{ji} + I_{(13)}^{ji} + I_{(17)}^{ji})}{\sum_{j=1}^n \sum_{x_j} \sum_{x_j^*} \tau_{x_j x_j^*}^{(s)} \cdot (I_{(02)}^{ji} + 2I_{(03)}^{ji} + I_{(04)}^{ji} + 2I_{(07)}^{ji} + I_{(12)}^{ji} + I_{(14)}^{ji} + I_{(16)}^{ji} + I_{(18)}^{ji} + I_{(26)}^{ji} + 2I_{(23)}^{ji} + I_{(28)}^{ji} + 2I_{(27)}^{ji} + 2 \cdot I_{(13)}^{ji} + 2 \cdot I_{(17)}^{ji})}.$$

### Updating θ

As for the third part of $Q(\Omega|\widetilde{\mathbf{X}}, \mathbf{Y}, \Omega^{(s)})$ in Eq. (2),

$$\sum_{j=1}^n \sum_{X_j} \log P(\widetilde{X}_j|X_j, \Omega) \cdot \tau_{X_j}^{(s)} = \sum_{j=1}^n \sum_{X_j} \log P(\widetilde{X}_j|X_j, \Omega) \cdot P(X_j = x_j|\widetilde{X}_j, Y_j, \Omega^{(s)}),$$

and maximization of this function could lead to an explicit expression of the error rate, $\theta$:

$$\theta^{(s+1)} = \frac{\sum_{j=1}^n \sum_{x_j} k_j \cdot P(X_j = x_j|\widetilde{X}_j, Y_j, \Omega^{(s)})}{n \cdot (q+1)}.$$

By repeating the above entire updating process for $C$, $\sum_e$, $\gamma$, and $\theta$ until convergence, the parameter estimate of $\Omega$ can eventually be obtained. For convenience, this proposed method for mapping QTLs of multiple traits is referred to as MTMIM_e.

In our study, we suppose that the overall genotype error rate is in a given data set, which means that the genotype error rates for each locus are same. When considering multiple datasets with same error rate (*e.g.*, the genotype data in each dataset were obtained under the same environment), we can combine these datasets into one big dataset. At this time, the total error rate is still the fixed parameter, so the combined dataset can be regarded as a new dataset, and then can be analyzed by the method provided in this paper. All parameters including the genotype error rate can be estimated, and moreover, the accuracy of parameter estimation will be improved due to the increase of sample size. If these datasets have different genotype error rates (*e.g.*, $\theta_1$, $\theta_2$, $\theta_3$ ), we can also combine these data sets into one big dataset, but the estimating strategy in our method need to be adjusted. The parameters $C$, $\sum_e$, and $\gamma$ can be estimated by the whole data, but the genotype error rates need to be estimated by their respective dataset.

## SIMULATION STUDIES

To evaluate the performance of the proposed MTMIM_e method in this study and objectively compare it with the existing methods, MTMIM (*Joehanes, 2009*) and MTMIM-NEW (*Tong, Sun & Zhou, 2018*), extensive simulation studies were conducted in this section.

In our simulation design, two quantitative traits and two latent causal QTLs were considered. The two QTLs were respectively located in two equally spaced marker intervals that consisted of three markers on a chromosome. We randomly set certain genotype errors at the first marker in error rate $\theta$. The length of the marker interval was taken as ten cM, and the sample size was $N = 500$. A map distance of ten cM corresponds to a
**Table 2 The estimates of all parameters (except $\theta$) and the corresponding mean square errors (MSEs) with different error rates when h2 = 0.2.**

| | Parameter | True Value | $\theta = 0$ MTMIM | MTMIM-NEW | MTMIM_e | $\theta = 0.05$ MTMIM | MTMIM-NEW | MTMIM_e | $\theta = 0.1$ MTMIM | MTMIM-NEW | MTMIM_e |
|---|---|---|---|---|---|---|---|---|---|---|---|
| | $\gamma_{11}$ | 0.03 | 0.0437 | 0.0318 | 0.0292 | 0.0554 | 0.0308 | 0.0241 | 0.0573 | 0.0297 | 0.0254 |
| | | | (0.0228)[c] | (0.0117) | (0.0067) | (0.0334) | (0.0193) | (0.0071) | (0.0240) | (0.0176) | (0.0070) |
| | $\gamma_{21}$ | 0.04 | 0.0351 | 0.0455 | 0.0402 | 0.0360 | 0.0362 | 0.0384 | 0.0343 | 0.0347 | 0.0386 |
| | | | (0.0193) | (0.0113) | (0.0133) | (0.0184) | (0.0193) | (0.0127) | (0.0196) | (0.0185) | (0.0138) |
| | $\rho$ | 0.9 | 0.8778 | 0.8866 | 0.8824 | 0.8730 | 0.8821 | 0.8744 | 0.8570 | 0.8819 | 0.8574 |
| | | | (0.0689) | (0.0622) | (0.0675) | (0.0710) | (0.0669) | (0.1053) | (0.0673) | (0.1669) | (0.1402) |
| | $\sigma_{11}$ | 1 | 0.9648 | 0.9846 | 0.9805 | 0.9779 | 0.9796 | 0.9720 | 0.9773 | 0.9794 | 0.9559 |
| | | | (0.0716) | (0.0656) | (0.0690) | (0.0703) | (0.0704) | (0.1055) | (0.0688) | (0.1679) | (0.1355) |
| | $\sigma_{22}$ | 1 | 0.9606 | 0.9839 | 0.9799 | 0.9636 | 0.9790 | 0.9709 | 0.9690 | 0.9796 | 0.9518 |
| | | | (0.0746) | (0.0657) | (0.0714) | (0.0784) | (0.0710) | (0.1137) | (0.0742) | (0.1719) | (0.1578) |
| T1 Q1 | | $-0.1478$[a] | $-0.1780$ | $-0.1427$ | $-0.1440$ | $-0.2121$ | $-0.1396$ | $-0.1322$ | $-0.2009$ | $-0.1288$ | $-0.1459$ |
| | | | (0.1528) | (0.1292) | (0.1305) | (0.2169) | (0.2139) | (0.1389) | (0.2012) | (0.1946) | (0.1773) |
| | | $0.1182$[b] | 0.1230 | 0.1082 | 0.1159 | 0.1664 | 0.1026 | 0.1019 | 0.1596 | 0.0775 | 0.0966 |
| | | | (0.1539) | (0.1527) | (0.1588) | (0.2424) | (0.2226) | (0.1939) | (0.2719) | (0.1949) | (0.2229) |
| | Q2 | 0.1288 | 0.1608 | 0.1207 | 0.1251 | 0.1905 | 0.1224 | 0.0998 | 0.1783 | 0.1113 | 0.0971 |
| | | | (0.1678) | (0.1296) | (0.1258) | (0.2008) | (0.2104) | (0.1083) | (0.2160) | (0.1911) | (0.1108) |
| | | 0.1492 | 0.1218 | 0.1570 | 0.1564 | 0.1128 | 0.1592 | 0.1764 | 0.1101 | 0.1848 | 0.1881 |
| | | | (0.1667) | (0.1538) | (0.1441) | (0.2479) | (0.2247) | (0.1383) | (0.2620) | (0.1945) | (0.1403) |
| T2 Q1 | | $-0.1478$ | $-0.1700$ | $-0.1473$ | $-0.1425$ | $-0.2148$ | $-0.1466$ | $-0.1354$ | $-0.1945$ | $-0.1328$ | $-0.1476$ |
| | | | (0.1404) | (0.1292) | (0.1269) | (0.2251) | (0.2263) | (0.1487) | (0.2005) | (0.1930) | (0.1931) |
| | | 0.1135 | 0.1200 | 0.1084 | 0.1117 | 0.1838 | 0.1060 | 0.1000 | 0.1589 | 0.0820 | 0.0978 |
| | | | (0.1555) | (0.1521) | (0.1609) | (0.2477) | (0.2259) | (0.1914) | (0.2753) | (0.1993) | (0.2369) |
| | Q2 | 0.2593 | 0.2794 | 0.2578 | 0.2541 | 0.3195 | 0.2582 | 0.2312 | 0.2996 | 0.2454 | 0.2271 |
| | | | (0.1450) | (0.1289) | (0.1223) | (0.2107) | (0.2233) | (0.1116) | (0.2058) | (0.1876) | (0.1120) |
| | | 0 | $-0.0241$ | $-0.0049$ | 0.0036 | $-0.0484$ | 0.0046 | 0.0259 | $-0.0312$ | 0.0265 | 0.0353 |
| | | | (0.1611) | (0.1561) | (0.1457) | (0.2557) | (0.2281) | (0.1383) | (0.2612) | (0.1971) | (0.1377) |

**Notes:**
[a] Additive effect of QTL.
[b] Dominant effect of QTL.
[c] MSE of the estimates for each parameter.

recombination rate of 0.0906 according to the Haldane map function (*Ott, 1999*). To study the impact of genotype errors on the estimates of the recombination rate $\gamma$, the effect matrix $C$, and the covariance matrix $\Sigma_e$, simulated data were generated under different error rates ($\theta = 0$, 0.05, and 0.1); the true values of recombination rates were taken as $\gamma_{11} = 0.03$ and $\gamma_{21} = 0.04$, and the effect matrix $C$ was selected to make the heritability $h^2$ close to 0.05 and 0.2, respectively. For each group of parameters, we computed the estimates of all parameters using the above three methods. The entire process was repeated 1,000 times and the mean of the estimates for each parameter was computed. To evaluate the accuracy of the estimates obtained by the different methods, the mean square error (MSE) of each parameter estimate and the total mean (TM) of MSEs of all parameter estimates (except $\theta$) were also calculated (Tables 2–4; Figs. 1, 2).

**Table 3 The estimates of all parameters (except $\theta$) and the corresponding MSEs with differentheritabilities when $\theta = 0.05$.**

| | | True value | | $h^2 = 0.05$ | | | $h^2 = 0.2$ | | |
|---|---|---|---|---|---|---|---|---|---|
| | Parameter | $h^2 = 0.05$ | $h^2 = 0.2$ | MTMIM | MTMIM-NEW | MTMIM_e | MTMIM | MTMIM-NEW | MTMIM_e |
| | $\gamma_{11}$ | 0.03 | 0.03 | 0.0560 | 0.0256 | 0.0191 | 0.0554 | 0.0308 | 0.0241 |
| | | | | (0.0340)[c] | (0.0250) | (0.0129) | (0.0334) | (0.0193) | (0.0071) |
| | $\gamma_{21}$ | 0.04 | 0.04 | 0.0364 | 0.0308 | 0.0343 | 0.0360 | 0.0362 | 0.0384 |
| | | | | (0.0239) | (0.0292) | (0.0188) | (0.0184) | (0.0193) | (0.0127) |
| | $\rho$ | 0.9 | 0.9 | 0.8762 | 0.8828 | 0.8714 | 0.8730 | 0.8821 | 0.8744 |
| | | | | (0.0773) | (0.0761) | (0.1128) | (0.0710) | (0.0669) | (0.1053) |
| | $\sigma_{11}$ | 1 | 1 | 0.9760 | 0.9824 | 0.9704 | 0.9779 | 0.9796 | 0.9720 |
| | | | | (0.0822) | (0.0700) | (0.1177) | (0.0703) | (0.0704) | (0.1055) |
| | $\sigma_{22}$ | 1 | 1 | 0.9722 | 0.9793 | 0.9663 | 0.9636 | 0.9790 | 0.9709 |
| | | | | (0.0831) | (0.0804) | (0.1226) | (0.0784) | (0.0710) | (0.1137) |
| T1 | Q1 | −0.0678[a] | −0.1478 | −0.0377 | −0.0676 | −0.0698 | −0.2121 | −0.1396 | −0.1322 |
| | | | | (0.2273) | (0.2166) | (0.1847) | (0.2169) | (0.2139) | (0.1389) |
| | | 0.0542 | 0.1182[b] | 0.0799 | 0.0616 | 0.0549 | 0.1664 | 0.1026 | 0.1019 |
| | | | | (0.2499) | (0.2349) | (0.1899) | (0.2424) | (0.2226) | (0.1939) |
| | Q2 | 0.0932 | 0.1288 | 0.0873 | 0.0946 | 0.0801 | 0.1905 | 0.1224 | 0.0998 |
| | | | | (0.2178) | (0.2120) | (0.1267) | (0.2008) | (0.2104) | (0.1083) |
| | | −0.0510 | 0.1492 | −0.0578 | −0.0524 | −0.0353 | 0.1128 | 0.1592 | 0.1764 |
| | | | | (0.2367) | (0.2291) | (0.1443) | (0.2479) | (0.2247) | (0.1383) |
| T2 | Q1 | −0.0678 | −0.1478 | −0.0307 | −0.0661 | −0.0700 | −0.2148 | −0.1466 | −0.1354 |
| | | | | (0.2164) | (0.2085) | (0.1883) | (0.2251) | (0.2263) | (0.1487) |
| | | 0.0520 | 0.1135 | 0.0792 | 0.0573 | 0.0492 | 0.1838 | 0.1060 | 0.1000 |
| | | | | (0.2490) | (0.2340) | (0.1908) | (0.2477) | (0.2259) | (0.1914) |
| | Q2 | 0.1089 | 0.2593 | 0.0895 | 0.1085 | 0.0953 | 0.3195 | 0.2582 | 0.2312 |
| | | | | (0.2079) | (0.2026) | (0.1261) | (0.2107) | (0.2233) | (0.1116) |
| | | 0 | 0 | −0.0122 | 0.0014 | 0.0168 | −0.0484 | 0.0046 | 0.0259 |
| | | | | (0.2405) | (0.2296) | (0.1463) | (0.2557) | (0.2281) | (0.1383) |

**Notes:**
[a] Additive effect of QTL when heritability $h^2 = 0.05$.
[b] Dominant effect of QTL when heritability $h^2 = 0.2$.
[c] MSE of the estimate for each parameter.

Table 2 provides the estimates of all parameters (except $\theta$) and the MSEs with different error rates when heritability $h^2 = 0.2$. Simulation results show that genotype error is an important factor affecting parameter estimation; larger genotype error rates would result in lower estimation accuracy. To see the performance of the three methods more intuitively, we present the TM histograms of the three methods (Fig. 1). The gray bar graph shows the TM values of the newly proposed method under different error rates. In the case of the same genotype error rate ($\theta = 0.05, 0.1$), it can be seen from Fig. 1 that each value of the TM of the proposed method (MTMIM_e) is uniformly lower than the corresponding values of MTMIM-NEW and MTMIM. When the genotype error rate $\theta = 0$ (*i.e.*, no genotype error in the simulated data set), the performance of the new method is also comparable to that of the MTMIM-NEW method and better than that of the

**Table 4 The estimates of error rate q with different heritabilities and QTL effects.**

| Heritability | Trait | QTL | Effect parameter | | True value of error rate $\theta$ | | |
|---|---|---|---|---|---|---|---|
| | | | Additive | Dominant | 0 | 0.05 | 0.1 |
| $h^2 = 0.2$ | T1 | Q1 | −0.1478 | 0.1182 | 0.0106 | 0.0585 | 0.1013 |
| | | Q2 | 0.1288 | 0.1492 | (0.0225) | (0.0703) | (0.1051) |
| | T2 | Q1 | −0.1478 | 0.1135 | | | |
| | | Q2 | 0.2593 | 0 | | | |
| $h^2 = 0.05$ | T1 | Q1 | −0.0678 | 0.0542 | 0.0306 | 0.0580 | 0.1019 |
| | | Q2 | 0.0932 | −0.0510 | (0.1296) | (0.1319) | (0.1369) |
| | T2 | Q1 | −0.0678 | 0.0520 | | | |
| | | Q2 | 0.1089 | 0 | | | |

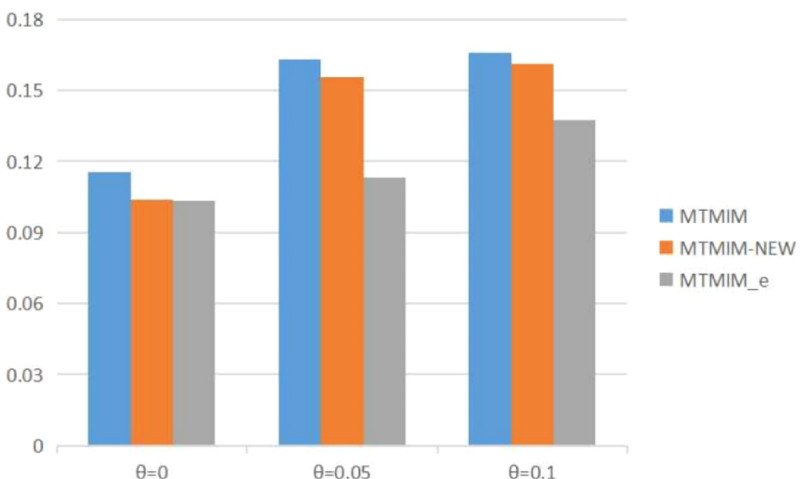

**Figure 1 The total means (TMs) of MSEs of all parameter estimates (except $\theta$) in Table 2.**

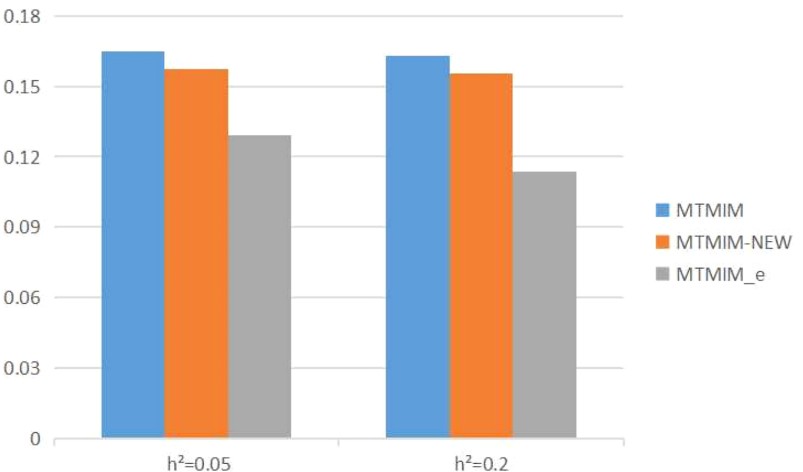

**Figure 2 The total means (TMs) of MSEs of all parameter estimates (except $\theta$) in Table 3.**

original MTMIM method. That is, the new method outperforms the other two methods in terms of estimating accuracy, and it can effectively overcome the influence of the genotype errors on gene mapping.

We also consider the impact of different heritabilities on the estimation accuracy. Table 3 gives the estimates of all parameters (except $\theta$) and their MSEs, with different heritability values when the genotype error rate $\theta = 0.05$. The simulation results showed that, with an increase in heritability, the accuracy of these three methods also increases as expected; however, among all these methods, the new method still has the highest estimation accuracy. Figure 2 shows the TM histogram of the three methods under different heritability ($h^2 = 0.05, 0.2$) conditions. The TM value of the new method (MTMIM_e) is smaller than that of the other two methods; in other words, the new method can weaken the influence of genotype error on parameter estimation (Fig. 2).

In addition to the estimates of QTL effects, QTL positions, and the covariance matrix of multiple traits, the estimate of the genotype error rate can also be obtained simultaneously using the new method. Table 4 shows the estimates of error parameter $\theta$ and the corresponding MSEs with true values of genotype error rate $\theta = 0, 0.05, 0.1$, when the heritability is $h^2 = 0.05, 0.2$. The estimation accuracy of error rate $\theta$ increases with an increase in heritability $h^2$, as expected, and the estimation accuracy decreases with an increase in the genotype error rate (Table 4). This is because simulated data with higher genotype error rates provide more uncertainty to the practical inference, relative to the cases with lower genotype errors or no genotype error.

In order to further evaluate the performance of the proposed method on analyzing multiple datasets, we conducted another simulation study. A total of three datasets with sample sizes $N_1 = 200$, $N_2 = 300$, and $N_3 = 500$, and genotype error rates $\theta_1 = 0.01$, $\theta_2 = 0.02$, and $\theta_3 = 0.05$ were simulated under two heritabilities (see Table 5 for the true values of parameters), and the MTMIM_e method was used to estimate all parameters. The means of the estimates for each parameter over 1,000 replications and the corresponding MSEs were provided in Table 5. From the results, we conclude that the new method can effectively and simultaneously analyze multiple datasets with potential genotype errors.

## STATISTICAL ANALYSIS OF REAL DATA

In this section, we further verified the feasibility of the proposed method in the mapping of multiple-trait loci, and an experimental data set on high-density lipoprotein cholesterol levels in mice in the published literature was used for the analysis (*Stylianou et al., 2006*). High-density lipoprotein (HDL) and body weight (BW) traits were selected for data analysis. We chose three candidate markers (D08.0989, D08.1099, and D08.1238) to perform multiple-interval mapping, which were located at 49.225, 54.685, and 61.605 cM of chromosome 8, respectively. These markers construct two marker intervals. In fact, there are missing values in the genotype data of the first marker of the considered dataset. For illustration, the missing values were imputed by the corresponding modes of genotypes on the locus, and then the new genotype data became observed data but with genotype errors.
**Table 5 The estimates of all parameters and the corresponding MSEs from three datasets.**

| Heritability | Trait | QTL | Effect parameter Additive | Dominant | Estimates Additive | Dominant | $\gamma_{11}$ 0.03 | $\gamma_{21}$ 0.04 | $\rho$ 0.9 | $\sigma_{11}$ 1 | $\sigma_{21}$ 1 | Error rate $\theta$ $\theta_1 = 0.01$ | $\theta_2 = 0.02$ | $\theta_3 = 0.05$ |
|---|---|---|---|---|---|---|---|---|---|---|---|---|---|---|
| $h^2 = 0.2$ | T1 | Q1 | −0.1478 | 0.1182 | −0.1457 | 0.1013 | 0.0269 | 0.0379 | 0.8844 | 0.9822 | 0.9822 | 0.0189 | 0.0301 | 0.0397 |
| | | | | | (0.1244) | (0.0929) | (0.0039) | (0.0076) | (0.0894) | (0.0944) | (0.0907) | (0.0751) | (0.0823) | (0.0814) |
| | | Q2 | 0.1288 | 0.1492 | 0.1120 | 0.1646 | | | | | | | | |
| | | | | | (0.0700) | (0.0972) | | | | | | | | |
| | T2 | Q1 | −0.1478 | 0.1135 | −0.1473 | 0.0977 | | | | | | | | |
| | | | | | (0.1253) | (0.0917) | | | | | | | | |
| | | Q2 | 0.2593 | 0 | 0.2436 | 0.0175 | | | | | | | | |
| | | | | | (0.0723) | (0.0985) | | | | | | | | |
| $h^2 = 0.05$ | T1 | Q1 | −0.0678 | 0.0542 | −0.0725 | 0.0492 | 0.0300 | 0.0413 | 0.8798 | 0.9779 | 0.9788 | 0.0263 | 0.0378 | 0.0374 |
| | | | | | (0.1350) | (0.1748) | (0.0040) | (0.0064) | (0.0962) | (0.1088) | (0.0926) | (0.0861) | (0.0932) | (0.0898) |
| | | Q2 | 0.0932 | −0.0510 | 0.0828 | −0.0465 | | | | | | | | |
| | | | | | (0.0774) | (0.0948) | | | | | | | | |
| | T2 | Q1 | −0.0678 | 0.0520 | −0.0699 | 0.0433 | | | | | | | | |
| | | | | | (0.1276) | (0.1521) | | | | | | | | |
| | | Q2 | 0.1089 | 0 | 0.0969 | 0.0064 | | | | | | | | |
| | | | | | (0.0782) | (0.0941) | | | | | | | | |

**Table 6 The QTLs identified in the high-density lipoprotein data of mice.**

| Trait | Chr. | QTL | MTMIM Position | Additive | Dominant | MTMIM-NEW Position | Additive | Dominant | MTMIM_e Position | Additive | Dominant |
|---|---|---|---|---|---|---|---|---|---|---|---|
| BW[a] | 8 | QTL1 | 49.7[c] | −0.0260 | 0.0037 | 50.2 | −0.0138 | −0.0089 | 50.4 | −0.0120 | −0.0087 |
| | | QTL2 | 60 | 0.0349 | −0.0172 | 56 | 0.0088 | −0.0042 | 56.8 | 0.0086 | −0.0093 |
| HDL[b] | 8 | QTL1 | 49.7 | 0.0262 | 0.0037 | 50.2 | 0.0190 | −0.0087 | 50.4 | 0.0203 | −0.0092 |
| | | QTL2 | 60 | 0.0394 | −0.0119 | 56 | 0.0139 | −0.0105 | 56.8 | 0.0110 | −0.0109 |

Notes:
[a] Body weight.
[b] High-density lipoprotein.
[c] The unit of QTL position is cM.

The proposed method, MTMIM_e, in which genotype errors are considered, and the other two methods, MTMIM and MTMIM-NEW, in which genotype errors are neglected, were all used to deal with the imputed data set (Table 6). All the estimated results of QTL positions and effects obtained from the three methods show that QTL 1 located at ~50 cM and QTL 2 located at ~57 cM of chromosome 8 had significant effects on the two traits, HDL and BW. Moreover, QTL 1 and QTL 2 showed additive/dominant effects on the two traits in the same direction for all three methods. A small difference is that the MTMIM method gave a positive dominant effect estimate on QTL 1, but the other two methods obtained a negative estimate value of the dominant effect of this QTL. These conclusions are similar to those drawn by Stylianou et al. (2006); however, the estimated results of MTMIM-NEW and MTMIM_e seem much closer to those obtained

by *Stylianou et al. (2006)*. In addition, the proposed MTMIM_e can simultaneously estimate the genotype error rate $\theta$ of the imputed data set, and the estimated value is $\hat{\theta} = 0.0252$, which is close to the missing proportion of genotypes on the first marker locus.

## DISCUSSION

In this paper, a new multiple-trait, multiple-interval mapping method (MTMIM_e) was proposed to deal with multiple-trait genetic data with potential genotype errors, owing to the fact that most genetic datasets may contain certain genotype measurement errors. In fact, the proposed method can also be used to analyze a dataset that contains missing values. In this case, one needs to impute the missing values by statistical methods first, and then consider the imputed missing values as data with genotyping errors, so that the MTMIM_e can be applied, as shown in our analysis of a real example. Extensive simulation studies have validated that the new method is advantageous for parameter estimation in the QTL mapping of multiple traits and it can effectively overcome the impact of genotype errors/loss and estimate all parameters simultaneously, compared with existing methods. The proposed method can also be used to deal with the problems of other cases, for example, other experimental populations, or different error rates on different loci, in which we only need to change the conditional probabilities provided in Table 1, or adjust the presentation of the joint error rate $\varphi^j$ (*Tong et al., 2015*), respectively. The statistical Model (1) considered in this study can also be further generalized; for example, genotype information of markers can be added and considered simultaneously, which will lead to a composite multi-interval mapping model.

After obtaining an estimate of $\Omega$, we can further consider whether there are significant QTLs in the considered marker intervals that control multiple traits. A global test using the likelihood-ratio (or log10 of odds; LOD score) statistic can be performed first to test two hypotheses of the effect matrix $C$, *i.e.*, $H_0: C = 0$ vs. $H_1 : C \neq 0$; and furthermore, a local test can be conducted to test two hypotheses about the specific QTL effect, *i.e.*, $H_0^{il} : C_{il} = 0$; $H_1^{il} : C_{il} \neq 0$, if the global test is significant. It is considered that there exists a significant QTL effect on trait $l$ in the considered marker interval if $H_0^{il} : C_{il} = 0$ is rejected.

Although the new method has several advantages, there are still aspects that can be improved. One disadvantage of the new method is that the amount of computation increases as the number of markers (intervals) or traits increases. For the computation of one simulated data containing 500 individuals (considering three maker loci and two phenotypes), the EM algorithm takes about 300 iterations to converge in the Windows System with Intel Core i5-10210U 2.11 GHz processor and 16 GB memory, and the memory consumption is about 500 MB. Phenotype amount has a little impact on memory consumption, but the most significant impact factor is the number of marker loci. When dealing with case of multiple real QTLs (or multiple markers), we suggest that two marker intervals constructed by three markers are analyzed by our method at one time. Performing this operation until all markers are completely analyzed will save much running time of the algorithm. Alternatively, we suggest using the idea of a two-step

mapping method (*Tong et al., 2015*), *i.e.*, first detect and retain the markers with larger effects of all the markers. Marker intervals can then be constructed using the selected markers so that the proposed method in this paper can be used with less computational burden. The problem of gene mapping is very important for human disease research, as well as for animal and plant genetic breeding; however, incomplete marker genotypes caused by biological or physical deletions are inevitable. In the future, we will focus on improving the performance of the proposed method in this paper, so that it can adapt better to more complex cases.

## ACKNOWLEDGEMENTS

The authors would like to thank the joint editor and referees for their helpful comments that greatly improved the presentation of the paper.

### Funding
This research was supported by the National Natural Science Foundation of China (Grant No. 12071114), the Natural Science Foundation of Heilongjiang Province of China (LH2019A020 and LH2019D018), the Fundamental Research Funds on Basic Research Project of Heilongjiang Provincial Colleges and Universities (KYYWF10236190103), and the Innovation Project for University Students of Heilongjiang Province (201910236004). The funders had no role in study design, data collection and analysis, decision to publish, or preparation of the manuscript.

### Grant Disclosures
The following grant information was disclosed by the authors:
National Natural Science Foundation of China: 12071114.
Natural Science Foundation of Heilongjiang Province of China: LH2019A020 and LH2019D018.
Heilongjiang Provincial Colleges and Universities: KYYWF10236190103.
University Students of Heilongjiang Province: 201910236004.

### Competing Interests
The authors declare that they have no competing interests.

### Author Contributions
- Liang Tong conceived and designed the experiments, prepared figures and/or tables, and approved the final draft.
- Ying Zhou conceived and designed the experiments, authored or reviewed drafts of the paper, and approved the final draft.
- Yixing Guo performed the experiments, prepared figures and/or tables, and approved the final draft.
- Hui Ding performed the experiments, prepared figures and/or tables, and approved the final draft.

- Donghai Ji analyzed the data, authored or reviewed drafts of the paper, and approved the final draft.

## Data Availability

The raw measurements are available in the Supplemental File.

## Supplemental Information

Supplemental information for this article can be found online at http://dx.doi.org/10.7717/peerj.12187#supplemental-information.

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
