# Peer review of "Quantitative trait locus mapping analysis of multiple traits when using genotype data with potential errors"

_PeerJ, doi:10.7717/peerj.12187_

## Round 0.1 · original submission · Minor Revisions

All reviewers recommended essential minor changes and I would like to invite the authors to revise their manuscript accordingly. Specifically, the language used in the manuscript needs to be improved therefore I recommend the authors consider professional English editing services.

·

Basic reporting

The text of the article is very neat and professional.

Experimental design

The design of simulated data can benefit from more extended explanation into its biological validity.

Validity of the findings

As pointed in general comments, the real test set needs to be expanded and a more rigorous benchmarking needs to be carried out against other existing methods.

Additional comments

In this article, the authors present a minor upgrade on previous a method on multi-trait QTL mapping. The authors first provide a rigorous mathematical definition of the problem, then provide a solution using EM algorithm. Then, first by using a set of simulated data then using a locus of real genotype data the method is being tested.

While the proposed method certainly is of real importance, I have a few questions and concerns that I think would help the manuscript to be more digestible for a reader:
1 – In the simulation studies part, presenting the results in terms of a large table is not advised. A graphical display followed with a statistical method such as ANOVA on the estimated parameters compared to the real values can be provided, at least as a supplementary figure. It is very hard to understand, from the table, the spread of deviations for different methods.
2 – The method needs to be run on multiple real QTL values from previously reported methods and benchmark against them. Using a single locus with 3 markers does not provide a sufficient amount of confidence in the performance evaluation.
3 – The study also needs to provide information on technical sides of the method such as; how many iteration EM takes to converge, memory consumption, estimated upper limits in terms of number of traints etc.

Reviewer 2 ·

Basic reporting

The article titled “QTL mapping analysis of multiple traits when using genotype data with potential errors (#62130)” by Tong L et al., proposed a new method of multiple trait and multiple interval mapping in the genetic data set with genotype errors.

Experimental design

Overall, the manuscript draft is well written and presented with good mathematical models and formulas. The statistical models are also justified.

Validity of the findings

The findings were of interest to the peers working around QTL mapping and multiple trait analysis and related studies.
The study design justifies the rationale of this work and the models were well designed. The studies were validated with statistical models and the limitations like excess amount of computation is also mentioned.

Additional comments

. As a reviewer from a reader’s point of view the below questions were made to clarify and improve the overall reach of this work.
Comments:
1. In general, what is the overall percentage of genotype errors in a given data set? Authors may estimate the % of errors from 3 or more large data sets and estimate the genotype errors in the data.
2. A table can be included with analysis of 3-5 data sets in which % genotype error can be determined and how successful is this newly proposed method can identify and fill the gaps
3. Can this method be applied to large data sets including human genome data, bacterial genomes data etc.? or is there a limitation for this method?
4. Can a computational tool/ application be developed for this method so that it can be of use for the scientists working in genome studies and other genomics works? If there is an existing website/software for the same, a link can be included in the discussion part.
Minor changes:
1. Line 19: Change to “in a given data set”
2. Line 20: Change to “limitations of biotechnology”
3. Line 51: Change to “on a mixed linear model”
4. Line 53: Change to “with the single-trait”
5. Line 185: Change to “The analyzed results”
6. Line 185: Change to “All the estimated results”
7. Line 189: Change to “gave a positive”
8. Line 190: Change to “obtained a negative”
9. Line 190: Change to “value of the dominant”
10. Line 199: Change to “analyze a data set”
11. Line 212: Change to “obtaining an estimate”
12. Line 217: Change to “QTL effect on”
13. Line 226: Change to “adapt to much”

Reviewer 3 ·

Basic reporting

In this manuscript the authors proposed a new QTL mapping strategy of multiple traits in the presence of the genotype errors in the genetic datasets. Within the new method, many parameters such as QTL recombination rates, covariance matrix, QTL effects and genotype error rates can be estimated, simultaneously. In order to show the accuracy of parameter estimation, they used simulated datasets which are generated under different scenarios and a real dataset, as well. Their results showed that their method can map QTLs of multiple traits more accurately than existing methods in the literature. The study is quiet interesting and the findings seem significant enough. I only have a few comments. Even though the main idea is clear I think the language should be improved for the intelligibility of the manuscript. There are some syntax and grammatical errors such as “i,e,....” in the line 214 and some punctuation errors such as “....; The true values “ in the line 143. Please do write the vectors in the formulas in bold. In the line 120, what does “G_....” refers to? In line 127, instead of using “.... above equation” give a number to the equation and refer it as Eq. ().

Experimental design

The research question is well defined. The experimental design, the theory and the statistical model are explained in details.

Validity of the findings

In order to show the accuracy of parameter estimation, they generated simulated datasets and applied their strategy on simulated datasets under different scenarios. Additionally, they performed the same analysis on a real data. The results seem to be robust, scientifically sound and support the authors in their findings.

Additional comments

In order to have a clearer manuscript, improve the readability of the manuscript and to reach out more researchers, I suggest the authors to make sure that they carefully read their manuscript from the scratch and make corrections accordingly. Please use a simple and clear language so that it does not lead to any confusion while reading the paper.

---

## Round 0.2 · accepted · Accept

The authors satisfactorily addressed the reviewers' comments. The manuscript can be now accepted for publication.

·

Basic reporting

No comment

Experimental design

Adequately explained in revision.

Validity of the findings

Might be of value for very specific tasks.

Additional comments

All the questions and comments that we posed were adequately addressed by the authors.

Reviewer 3 ·

Basic reporting

None.

Experimental design

None

Validity of the findings

None

Additional comments

The authors have answered my comments and I have nothing else to add.